# Transcriptional Suppression of Diabetic Nephropathy with Novel Gene Silencer Pyrrole-Imidazole Polyamides Preventing USF1 Binding to the TGF-β1 Promoter

**DOI:** 10.3390/ijms22094741

**Published:** 2021-04-29

**Authors:** Makiyo Okamura, Noboru Fukuda, Shu Horikoshi, Hiroki Kobayashi, Akiko Tsunemi, Yurie Akiya, Morito Endo, Taro Matsumoto, Masanori Abe

**Affiliations:** 1Division of Nephrology, Hypertension and Endocrinology, Department of Medicine, Nihon University School of Medicine, Ooyaguchi-kami 30-1, Itabashi-ku, Tokyo 173-8610, Japan; makiyo91supika@yahoo.co.jp (M.O.); horiko19830823@yahoo.co.jp (S.H.); kobayashihiroki2@gmail.com (H.K.); tsunemi.akiko@nihon-u.ac.jp (A.T.); akiya.yurie@nihon-u.ac.jp (Y.A.); 2Division of Cell Regeneration and Transplantation, Department of Functional Morphology, Nihon University School of Medicine, Tokyo 173-8610, Japan; matsumoto.taro@nihon-u.ac.jp; 3Faculty of Human Health Science, Hachinohe Gakuin University, Hachinohe, Aomori 031-8588, Japan; mendo@hachinohe-u.ac.jp

**Keywords:** diabetic nephropathy, osteopontin, pyrrole-imidazole polyamide, rat, TGF-β1, USF1

## Abstract

Upstream stimulatory factor 1 (USF1) is a transcription factor that is increased in high-glucose conditions and activates the transforming growth factor (TGF)-β1 promoter. We examined the effects of synthetic pyrrole-imidazole (PI) polyamides in preventing USF1 binding on the TGF-β1 promoter in Wistar rats in which diabetic nephropathy was established by intravenous administration of streptozotocin (STZ). High glucose induced nuclear localization of USF1 in cultured mesangial cells (MCs). In MCs with high glucose, USF1 PI polyamide significantly inhibited increases in promoter activity of TGF-β1 and expression of TGF-β1 mRNA and protein, whereas it significantly decreased the expression of osteopontin and increased that of *h*-caldesmon mRNA. We also examined the effects of USF1 PI polyamide on diabetic nephropathy. Intraperitoneal injection of USF1 PI polyamide significantly suppressed urinary albumin excretion and decreased serum urea nitrogen in the STZ-diabetic rats. USF1 PI polyamide significantly decreased the glomerular injury score and tubular injury score in the STZ-diabetic rats. It also suppressed the immunostaining of TGF-β1 in the glomerulus and proximal tubules and significantly decreased the expression of TGF-β1 protein from kidney in these rats. These findings indicate that synthetic USF1 PI polyamide could potentially be a practical medicine for diabetic nephropathy.

## 1. Introduction

As diabetic nephropathy is the most frequent cause of renal failure in Japan, the drug discovery of novel radical medicines is required to suppress the progression of renal dysfunction in diabetes mellitus. In its early stage, diabetes mellitus induces podocyte injury with endothelial damage in the glomerulus that causes the loss of glomerular barrier function. Subsequently, diabetes mellitus induces glomerular sclerosis with mesangial cell (MC) proliferation [1].

Transforming growth factor (TGF)-β is a responsible factor for degeneration of the kidney in renal diseases such as glomerulosclerosis, mesangioproliferative glomerulonephritis [2] and diabetic nephropathy. High glucose levels induce TGF-β1 gene expression that causes glomerular sclerosis and interstitial fibrosis with the synthesis of extracellular matrices (ECMs) and epithelial–mesenchymal transition (EMT) of nephrotubules [3]. Increased TGF-β1 further induces endothelial–mesenchymal transition (EndMT) to impair glomerular endothelial function, by which the failure of crosstalk between glomerular endothelial cells and podocytes causes proteinuria [4]. Thus, TGF-β1 is a responsible molecule involved in the pathogenesis of diabetic nephropathy.

Pyrrole-imidazole (PI) polyamides can form specific hydrogen bonds to double-stranded DNA (dsDNA) [5], which is stronger than the protein bonding to dsDNA. dsDNA recognition is based on the pairing of pyrrole (Py) with imidazole (Im) in the minor groove. This Im-Py pairing targets the G-C base pair, whereas the opposite pairing (Py-Im) targets the C-G base pair. The pairing of Py-Py targets the A-T and T-A base pairs [6]. Peptide compound PI polyamides show resistance to nuclease and are delivered into cells and nuclei without the need of a drug delivery system. We previously demonstrated that PI polyamide injected intravenously distributed into the kidney and aorta, but did not distribute considerably in he art or brain in rats [7]. Thus, synthetic PI polyamides designed to be promoters of a target should be effective practical medicines as gene silencers. We have shown that PI polyamides targeting TGF-β1 effectively improved renal sclerosis with hypertension [7,8] and diabetic nephropathy in rats [9] and progressive renal diseases in the primate common marmoset [10].

It has been reported that high-glucose stimulations activate protein kinase C and the polyol pathway that induce accumulation of radical oxygen species and advanced glycation endproducts [11]. These factors activate transcription factors such as upstream stimulatory factor, activated protein-1 (AP-1) and CREB, which facilitate the transcription of TGF-β1 and induce inflammation and fibrosis in kidney [12].

We recently examined the effects of PI polyamide targeting the AP-1 binding site of the TGF-β1 promoter on diabetic nephropathy in streptozotocin (STZ)-induced diabetes mellitus in rats, in which we did not find improvement of histological degeneration [9]. It is considered that the activation of the TGF-β1 promoter by AP-1 transcription factor binding is not specific for diabetic nephropathy, whereas the transcription factor USF1 is specifically activated by high-glucose stimulation in diabetes mellitus and activates the TGF-β1 promoter. Thus, PI polyamides that prevent USF1 binding on the TGF-β1 promoter are expected to be a novel specific medicine for diabetic nephropathy.

In the present study, we molecularly designed and synthesized multiple PI polyamides that prevent USF1 binding on the TGF-β1 promoter and examined both their effects on the promoter activity and mRNA expression of TGF-β1 in renal MCs in high-glucose conditions to determine the lead compound in vitro and the effects of injection of the lead compound on diabetic nephropathy in vivo in rats with STZ-induced diabetes mellitus.

## 2. Results

### 2.1. Gel Mobility Shift Assay and DNA Binding Assays for USF1 PI Polyamides

The bindings of four PI polyamides designed to span the boundary of USF1-binding E-box sequences on target dsDNAs were evaluated by gel mobility shift assay (Figure 1A). USF1 PI polyamide-1, -2, -3 and -4 bound the appropriate 21-bp FITC-labeled dsDNAs, which are indicated by the shift of bands (Lane 2). The addition of excessive non-FITC labeled dsDNAs showed both shifted and non-shifted bands (Lane 3). USF1 PI polyamide-1, -2, -3 and -4 did not bind mismatch dsDNAs. These findings indicate that the four synthesized PI polyamides can bind target dsDNA.

As shown in Figure 1B, a single mobility band was observed when the dsDNA was incubated with USF1 protein (Lane 2). Polyamide-3 completely inhibited USF1 binding to target dsDNA (Lane 4). Mismatch polyamide did not affect USF1 binding to target dsDNA (Lane 6).

### 2.2. Cellular Localization of USF1 Protein in MCs with High-Glucose Stimulation

Immunofluorescence microscopy showed that USF1 protein was localized in the cytoplasm and nuclei in MCs cultured with normal-glucose medium, whereas USF1 protein was translocated from cytoplasm to nuclei in MCs cultured with high-glucose medium (Figure 2A).

Figure 2B,C shows expression of USF1 protein in the cytoplasm and nuclei of MCs with normal-glucose or high-glucose stimulation by Western blot analysis. In MCs cultured with high-glucose stimulation, there was no difference in the abundance of USF1 protein in the cytoplasm but that of USF1 protein was significantly (*p* < 0.05) increased in the nuclei of the MCs. These findings indicate that a high-glucose condition stimulates the translocation of USF1 protein from the cytoplasm to nuclei of MCs.

### 2.3. Effects of USF1 PI Polyamides on TGF-β1 Promoter Activity with High-Glucose Stimulation

We examined the effects of USF1 PI polyamide-1, -2, -3 and -4 on TGF-β1 promoter activity measured by luciferase activity of TGF-β1 promoter plasmid transfected in HEK293 cells. High-glucose stimulation significantly (*p* < 0.05) increased the luciferase activity. A concentration of 10^−10^ M of USF1 PI polyamide-3 significantly (*p* < 0.05) decreased glucose-stimulated luciferase activity (Figure 3A), whereas USF1 PI polyamide-1, -2 and -4 did not suppress the increase in luciferase activity.

### 2.4. Effects of USF1 PI Polyamides on TGF-β1 Expression in MCs with High-Glucose Stimulation

The high-glucose condition increased the abundance of TGF-β1 mRNA in MCs. Concentrations of 10^−10^ to 10^−8^ M of USF1 PI polyamide-3 significantly (*p* < 0.01) inhibited increases in the abundance of TGF-β1 mRNA in MCs with high-glucose condition (Figure 3B), whereas USF1 PI polyamide-1, -2 and -4 did not suppress the increased abundance of TGF-β1 mRNA in MCs (Appendix A). Based on the results of experiments on the effects of USF1 PI polyamides on TGF-β1 promoter activity and mRNA expression, USF1 PI polyamide-3 was used for the following experiments.

A concentration of 10^−10^ M USF1 PI polyamide-3 significantly (*p* < 0.01) inhibited the increased abundance of TGF-β1 protein in MCs with high-glucose condition (Figure 3C). However, mismatch polyamide did not affect the abundance of TGF-β1 protein in MCs (Figure 3D).

### 2.5. Effects of USF1 PI Polyamide-3 on the Expression of Phenotype Markers in MCs with High-Glucose Stimulation

High-glucose stimulation significantly (*p* < 0.01) increased the abundance of osteopontin, a synthetic phenotype marker, mRNA but decreased the abundance of *h*-caldesmon, a contractile phenotype marker, mRNA in MCs (Figure 4A,B). A concentration of 10^−10^ M of USF1 PI polyamide-3 significantly (*p* < 0.01) inhibited the increased abundance of osteopontin in MCs with high-glucose conditions (Figure 4A). A concentration of 10^−11^ to 10^−8^ M of USF1 PI polyamide-3 significantly (*p* < 0.01) increased the inhibited abundance of *h*-caldesmon in MCs with high-glucose conditions (Figure 4B). These findings indicate that USF1 is involved in the phenotypic changes in MCs with high-glucose stimulation.

### 2.6. Delivery of PI Polyamide in Rat Kidney

Figure 5 shows the delivery of 2.5 mg/body of fluorescein isothiocyanate (FITC)-labeled PI polyamide-3 by intraperitoneal injection into rat kidney. USF1 PI polyamide-3 was mainly distributed into the nucleus of the nephron tubule, but was slightly distributed into the glomerulus, on Day 1. Distribution of USF1 PI polyamide-3 was observed mainly in the nucleus of the nephron tubule on Day 3.

### 2.7. Effects of USF1 PI Polyamide-3 on Body Weight, Urine Volume and Urinary Albumin Excretion in STZ-Diabetic Rats

We intraperitoneally injected 1 mg/kg/BW of USF1 PI polyamide twice a week into STZ-diabetic rats for 16 weeks. BW was lower in the STZ-diabetic rats than that in the control rats. Treatment with USF1 PI polyamide-3 did not affect the decrease in body weight in the STZ-diabetic rats (Figure 6A). Urine volume was markedly higher in the STZ-diabetic rats than that in the control rats. Treatment with USF1 PI polyamide-3 did not affect the increase in urine volume in the STZ-diabetic rats (Figure 6B). Urine volume was markedly higher in the STZ-diabetic rats than that in the control rats from 1 to 16 weeks. Urinary albumin excretion was higher in the STZ-diabetic rats than that in the control rats from 2 to 16 weeks. Treatment with USF1 PI polyamide-3 significantly (*p* < 0.05) decreased the increase in urinary albumin excretion in the STZ-diabetic rats (Figure 6C).

### 2.8. Effects of USF1 PI Polyamide-3 on Renal Function, Kidney Weight, Blood Sugar and HbA1c in STZ-Diabetic Rats

Levels of serum urea nitrogen (UN), kidney weight, blood sugar and HbA1c were significantly (*p* < 0.01) higher in the STZ-diabetic rats than those in the control rats at 16 weeks. Treatment with USF1 PI polyamide-3 significantly (*p* < 0.01) decreased the levels of serum UN in the STZ-diabetic rats (Figure 7A). However, it did not affect the increase in serum creatinine, kidney weight, blood sugar and HbA1c in the STZ-diabetic rats (Figure 7B–E).

### 2.9. Effects of USF1 PI Polyamide-3 on Kidney Degeneration in STZ-Diabetic Rats

Hematoxylin and eosin (HE) staining showed enlargement of the glomerulus with mesangial proliferation in kidney from the STZ-diabetic rats compared to the control rats. Moreover, the tubular interstitium showed fibrotic change and atrophy of nephrotubules as histology of diabetic nephropathy. Masson’s trichrome staining showed increases in fibrosis in kidney from the STZ-diabetic rats compared to the control rats (Figure 8A). The diameter of the glomerulus in the STZ-diabetic rats was significantly (*p* < 0.01) larger than that in the control rats. Treatments with USF1 PI polyamide-3 significantly (*p* < 0.05) suppressed enlargement of the glomerulus in kidney from the STZ-diabetic rats (Figure 8B). The glomerular injury score (GIS) and tubulointerstitial injury score (TIS) were significantly (both, *p* < 0.01) higher in kidney from the STZ-diabetic rats than those in the control rats. Treatments with USF1 PI polyamide-3 significantly (*p* < 0.05) inhibited increases in the GIS and TIS of kidney from the STZ-diabetic rats (Figure 8C,D).

### 2.10. Effects of USF1 PI Polyamide-3 on the Expression of TGF-β1 in STZ-Diabetic Rats

Images of the immunohistological staining of TGF-β1 in kidney from control and STZ-diabetic rats without or with USF1 PI polyamide-3 are shown in Figure 8A. TGF-β1 was strongly stained in the glomerulus and proximal tubules in kidney from STZ-diabetic rats compared to that from control rats. Treatments with USF1 PI polyamide-3 for 4 months significantly (*p* < 0.05) weaken the enhanced staining of TGF-β1 (Figure 9A). The expression of TGF-β1 in the renal cortex was evaluated by Western blot analysis. The abundance of TGF-β1 protein was significantly (*p* < 0.01) higher in the renal cortex of kidney from the STZ-diabetic rats compared to that from the control rats. Treatments with USF1 PI polyamide-3 for 4 months significantly (*p* < 0.01) decreased the staining of the abundant TGF-β1 (Figure 9B).

## 3. Discussion

It has been reported that USF1 is stimulated through the protein kinase-C/polyol pathway induced by high glucose in diabetic nephropathy [11]. Sanchez et al. [13] reported that the uptake of excessive glucose into cells inhibits AMPK activity and induces the nuclear localization of USF1 to stimulate transcription of TGF-β1 that causes progression to renal fibrosis. Thus, USF1 activation with high-glucose stimulation in diabetes mellitus enhances gene expression of TGF-β1 that induces an accumulation of ECM, inflammation and glomerular sclerosis in kidney. In the present study, we designed and synthesized PI polyamides to prevent USF1 binding on the rat TGF-β1 promoter to suppress expression of TGF-β1 and evaluated whether the PI polyamides could prevent the progression of diabetic nephropathy.

The contractile mesenchymal cells including MCs express contractile markers such as α-smooth muscle actin, smooth muscle 22α and *h*-caldesmon, whereas the synthetic mesenchymal cells showing enhanced growth with lower contractility express synthetic markers, such as osteopontin, matrix Gla and *l*-caldesmon. Osteopontin is an ECM protein found in bone marrow in which osteopontin is produced by osteoblast and mesenchymal stem cells [14]. Osteopontin is a potent molecule that induces the synthetic phenotype of mesenchymal cells and was reported to strongly correlate with urinary albumin excretion in diabetic nephropathy [15]. Thus, transcription factor USF1 is thought to bind on the CCTCATGAC motif of the TGF-β1 promoter to induce renal fibrosis, and it binds on the same CCTCATGAC motif of the osteopontin promoter to induce glomerular enlargement with the synthetic phenotype and proliferation of MCs in diabetic nephropathy.

Weigert et al. [16] showed that human TGF-β1 promoter has high homology to glucose response elements (GlRE) that are involved in glucose metabolism. They studied the function of GlRE in high glucose-induced TGF-β1 gene activation in MCs and found that high glucose enhances the nuclear amount of USF and its binding to the GlRE. We investigated the cellular localization of USF1 protein in MCs with high-glucose stimulation. Immunofluorescence microscopy showed that the USF1 protein was translocated from cytoplasm to nuclei in MCs cultured with high-glucose medium. Moreover, Western blot analysis showed that the expression of USF1 protein was significantly increased in the nuclei of MCs cultured with high glucose. These findings indicate that high glucose stimulates the translocation of USF1 protein from cytoplasm to the nuclei of MCs.

We designed four different PI polyamides to prevent USF1 binding on the E-box sequences of the rat TGF-β1 promoter and synthesized them by automatic solid-phase synthesis methods. We confirmed bindings of the synthesized USF1 PI polyamides on dsDNA by gel mobility shift assay. All four of the USF1 PI polyamides bound the target dsDNAs. However, only USF1 PI polyamide-3 significantly suppressed the increases in promoter activity and mRNA expression of TGF-β1 with high-glucose stimulation. USF1 PI polyamide-1, -2 and -4 did not suppress them. We therefore used USF1 PI polyamide-3 as the lead compound in the following in vitro and in vivo experiments to examine the effects of USF1 PI polyamide on the pathogenesis of diabetic nephropathy. In these experiments, high-glucose stimulation increased the expression of osteopontin, whereas it decreased the expression of *h*-caldesmon in MCs. USF1 PI polyamide-3 significantly inhibited the increased expression of osteopontin and stimulated the decreased expression of *h*-caldesmon in MCs. These findings suggest that USF1 PI polyamide-3 suppresses change in the synthetic phenotype of MCs with high glucose by blocking USF1 on the osteopontin promoter.

In the in vivo experiments, we examined the delivery of injected PI polyamide in rat kidney. USF1 PI polyamide-3 was distributed into the nucleus of the nephron tubule and glomerulus at 3 days after injection. Thus, USF1 PI polyamide-3 can distribute into the organs and strongly bind to their nuclei by themselves. This delivery property of the PI polyamide administered into the body could be a potential advantage as medicines, compared to that of nucleic acid medicines.

We created diabetic rats with the injection of STZ and examined the pharmacological effects of USF1 PI polyamide-3 on the pathogenesis of diabetic nephropathy. In the STZ-diabetic rats, water intake and urine volume were markedly increased by hyperglycemia, and body weight was decreased. Urinary albumin excretion was also increased from 12 to 16 weeks after the onset of diabetes. Kidney weight and serum UN were increased with glomerular enlargement in the diabetic rats. Histological evaluations by HE and Masson’s trichrome stainings showed glomerular enlargement with mesangial proliferation, fibrosis of the renal medulla and nephrotubular atrophy in kidney from the diabetic rats 16 weeks after the onset of diabetes. Immunohistological staining of TGF-β1 in kidney from the STZ-diabetic rats was enhanced in the glomerular mesangial region and proximal tubules. Treatment with intraperitoneal injections of USF1 PI polyamide-3 over 16 weeks significantly decreased the increase in urinary albumin excretion in the STZ-diabetic rats. TGF-β1 is known to induce both podocyte injury and EndMT by which endothelial function is impaired by mesenchymal transformation [17]. Thus, the increase in TGF-β1 contributes the impaired functions of glomerular endothelial cells and podocytes by disrupting their crosstalk in diabetic nephropathy.

Treatment with USF1 PI polyamide-3 significantly decreased levels of serum UN and improved the enlargement of the glomerulus in kidney from the STZ-diabetic rats. The GIS and TIS were significantly higher for kidney from the STZ-diabetic rats than the control rats, and treatments with USF1 PI polyamide-3 significantly decreased these scores for kidney from the STZ-diabetic rats. Treatments with USF1 PI polyamide-3 significantly weaken the enhanced staining of TGF-β1 in the glomerular mesangial region and proximal tubules in the STZ-diabetic rats. The treatments also decreased the protein expression of TGF-β1 in renal medulla in the STZ-diabetic rats. These findings indicate that long-term administration of USF1 PI polyamide-3 can improve impaired renal function and kidney degeneration by suppressing the increased expression of TGF-β1 in kidney from these rats.

The specificity of PI polyamides to target genes is important for drug discovery. We investigated the specificity of a PI polyamide targeting the AP-1 site of the TGF-β1 promoter. After systematic administration of the PI polyamide in Dhal salt-sensitive rats, we investigated its specificity by microarray analysis on the target transcript and found that the PI polyamide suppressed the target transcript in a specific manner [8]. USF1 PI polyamide-3 recognizes and binds eight base pair of dsDNAs, which can bind many genes rather than the TGF-β1, and possibly induces off-targeting effects. However, PI polyamides designed to prevent the binding of transcription factors only inhibit increases in promoter activity in the disease state; they do not inhibit the silencing of genes. Concerning the present USF1 PI polyamide-3, with high-glucose stimulation, the activity of USF1 is stimulated and translocated to the nucleus and binds the TGF-β1 promoter. Thus, it is possible that USF1 PI polyamide-3 specifically suppresses the TGF-β1 transcript rather than the TGF-β1 PI polyamides.

On the basis of these experiments, we are planning to design and synthesize human USF1 PI polyamides and examine their effects on renal degeneration in primate common marmoset injected with STZ, which has 90% homology with human genes. Then, we will develop USF1 PI polyamide-3 as a practical medicine for diabetic nephropathy.

Figure 10 presents a diagram of the involvement of USF1 in the pathogenesis of diabetic nephropathy and the transcriptional inhibition of TGF-β1 and osteopontin promoters by USF1 PI polyamide. Hyperglycemia induces translocation of USF1 from cytoplasm to nucleus in diabetes mellitus. Then, USF1 binds E-box sequences on both TGF-β1 and osteopontin promoters. The increased TGF-β1 stimulates EMT, EndMT and ECM formation, which induce renal fibrosis, glomerular endothelial damage, glomerular sclerosis and podocyte injury, and the increased osteopontin changes the phenotype of MCs from contractile to synthetic, which induces MC proliferation in the pathogenesis of diabetic nephropathy. Administration of USF1 PI polyamide will prevent the binding of USF1 on TGF-β1 and osteopontin promoters to reduce the pathogenesis of diabetic nephropathy.

## 4. Materials and Methods

### 4.1. Designing and Synthesizing PI Polyamides to Prevent USF1 Binding on the TGF-β1 Promoter

Appendix A shows sequences of the rat TGF-β1 promoter. Red color sequences are USF1-binding E-box regions. We designed four PI polyamides (polyamide-1, -2, -3 and -4) spanning the E-box sequences and TGF-β1 promoter sequence to obtain specificity for the TGF-β1 promoter (Figure 11).

We performed automatic solid-phase synthesis by first washing with dimethylformamide (DMF), then using 20% piperidine/DMF to remove the Fmoc group, followed by washing with methanol, coupling for 60 min with a monomer in a 1-(bis(dimethylamino)methylene)-5-chloro-1H-benzotriazolium 3-oxide hexafluorophosphate and diisopropylethylamine (4 eq each) environment, and performing another wash with methanol, protecting with acetic anhydride/pyridine, and, finally, another wash with DMF. After removal of the Fmoc group from the Fmoc-β-alanine-Wang resin, the resin was successively washed with methanol. The coupling step was performed with Fmoc-amino acid, followed by another methanol wash. These steps were repeated until all sequencing was complete. Thereafter, the N-terminal amino group was protected and washed with DMF, followed by draining of the reaction vessel. Next, after the cleavage step with 5 mL of 91% trifluoroacetic acid-3% triisopropylsilane-3% 5 dimethylsulfide-3% water/0.1 mmol resin, we used cold ethyl ether precipitation to isolate the synthetic polyamides. After another cleavage step (5 mL of N, N-dimethylaminopropylamine/0.1 mmol resin, 50 °C overnight), the synthetic polyamides were isolated again by cold ethyl ether precipitation. The polyamides were purified by HPLC using a PU-980 HPLC pump, UV-975 HPLC UV/VIS detector (Jasco, Easton, MD, USA), and Chemcobound 5-ODS-H column (Chemco Scientific, Osaka, Japan).

### 4.2. Gel Shift and DNA BindingAssays for PI Polyamides

To confirm that the designed and synthesized PI polyamides can bind target double-strand DNA, we performed gel shift assays. We synthesized sense oligonucleotide FITC-TGACTACTATGTGGAGTGGAT and antisense oligonucleotide FITC-ATCCACTCCACATAGTAGTCA for USF1 PI polyamides-1 and -2, and synthesized sense oligonucleotide FITC-CACTGCCACCAGTCACCATCA and antisense oligonucleotide FITC-TGATGGTGACTGGTGGCAGTG for USF1 PI polyamide-3 and -4. Two oligonucleotides (10^−5^ M) were denatured by heating at 95 °C for 5 min, followed by slow cooling so they could anneal with each other. The double-strand oligonucleotides and the same concentration of PI polyamide dissolved in dimethyl sulfoxide (DMSO, 07-4875-6, Sigma-Aldrich, St. Louis, MO, USA) were incubated in binding buffer (20% glycerol, 5 mM MgCl_2_, 2.5 mM EDTA, 250 mM NaCl, 50 mM Tris-HCl (pH 7.5)) for 2 h. 

For the DNA binding assay, one micromole of the dsDNA was incubated with 50 µM PI polyamide-3 or mismatched with or without USF1 protein (Proteinteck Ag17789, Rosemont, IL, USA) for 1 h at 37 °C. Those resulting complexes were separated by electrophoresis and visualized by luminescent image analyzer LAS-3000 (Fujifilm, Tokyo, Japan). 

### 4.3. Immunofluorescence Staining of USF1 in MCs

MCs were inoculated and cultured on a 35 mm glass base dish (IWAKI 3910-035, Tokyo, Japan) in DMEM with 10% FBS and normal glucose. After serum starvation in DMEM with 0.5% FBS for 24 h, cells were incubated in DMEM with 0.5% FBS and normal glucose or high glucose for 20 h. Then, cells were fixed with 4% *paraformaldehyde for 10 min and* permeabilized in PBS with 0.25% Triton-X for 15 min. After blocking with 10% albumin for 20 min, cells were incubated with rabbit anti-USF1 antibody (ab180717, Abcam, Cambridge, UK) diluted to 1:100 for 1 h. After washing with PBS, cells were incubated with Alexa-594 Goat anti-Rabbit IgG (A11072, Invitrogen, Waltham, MA, USA) as the second antibody for 30 min. Nuclei in the cells were stained with Hoechst 33342 (H1399, Invitrogen) diluted 1:1000.

### 4.4. Rat TGF-β1 Promoter Assay and Effects of USF1 Polyamides on Promoter Activity

Rat TGF-β1 promoter cloned into the KpnI site of the pGL3-basic vector was a gift from Dr. Kyoung Lim (Ajou University School of Medicine, Suwon, Korea). Human Embryonic Kidney cells 293 (HEK-293 cells) were obtained from the American Type Culture Collection (ATCC). Cells were seeded onto 24-well plates and grown in DMEM with 20% calf serum. At 70 to 90% confluence, a mixture of reporter plasmid (1 µg/well) and phRG-TK vector (0.01 µg/well; Promega, Madison, WI, USA) as an internal control was used to transfect cells with Lipofectamine 2000 (Invitrogen, Carlsbad, CA, USA) as described previously [18]. Cells were incubated for an additional 24 h and scraped into 100 µL of cold lysis buffer (PBS (pH 7.4) and 1 mM PMSF). Luciferase activity was measured with a Dual-luciferase reporter assay system (Promega) and a TD-20/20 luminometer (Turner Designs, Sunnyvale, CA, USA) [19]. For evaluation of the effect of USF1 polyamides on promoter activity, HEK-293 cells were transfected with rat TGF-β1 promoter plasmid and incubated with 0.1 or 1.0 µM polyamides in the presence of 25 mM glucose for 24 h. Luciferase activity then was measured.

### 4.5. Culture of MCs and Glucose Stimulation with USF1 Polyamides

The rat MC line ATCC CRL-2573 (ATCC) was maintained in RPMI 1640 with 10% fetal bovine serum (FBS) (Gibco, Waltham, MA, USA) and 0.05 mg/mL gentamicin (Gibco). The cells were inoculated on 6-well plates in normal glucose DMEM with 10% FBS and cultured until 80% confluency. Then, the medium was changed to DMEM with 0.5% FBS for 24 h to establish the serum starvation. Three hours before the end of serum starvation, USF1 polyamides were added into culture medium as final concentrations of 10^−10^ M, 10^−9^ M and 10^−8^ M in normal-glucose medium (glucose 5.6 mM (1000 mg/L) in DMEM) and then cultured with the polyamides in high-glucose medium (glucose 25 mM (4500 mg/L) in DMEM) for 24 h. Then, we performed real-time PCR analysis for TGF-β1 mRNA and Western blot analysis for TGF-β1, osteopontin and *h*-caldesmon proteins.

### 4.6. RNA Extraction and Real-Time PCR Analysis

Total RNA was extracted from the MCs with TRIzol reagent (Invitrogen, Carlsbad, CA, USA). A Takara RNA PCR Kit (AMV) version 3.0 (Takara Bio, Shiga, Ohtsu, Japan) was then used to reverse transcribe total RNA (1 μg) into cDNA with random 9-mers. Relative quantitation analysis of quantitative PCR (qPCR) was performed with Probe qPCR Mix (Takara Bio) and a Step-One Plus Real-Time PCR System (Applied Biosystems, Waltham, MA, USA). The TaqMan probes used in the qPCR were as follows: osteopontin (Rn00681031_ m1), USF1, TGF-β1, and *h-*caldesmon (Appendix A). We used glyceraldehyde-3-phosphate dehydrogenase (GAPDH) (4351317; Life Technologies, Carlsbad, CA, USA) with TaqMan Ribosomal RNA Control Reagents (Applied Biosystems) to control sample loading. The amplification conditions were 50 °C for 2 min, 95 °C for 10 min, 60 cycles of denaturation at 95 °C for 15 s, and combined annealing and extension at 60 °C for 1 min. After we determined the threshold cycle (Ct), we used the comparative Ct method to calculate the relative quantification of mRNA expression of the marker gene.

### 4.7. Western Blot Analysis

MCs in culture and renal medulla from rats were disrupted with lysis buffer (50 mM Tris·HCl, pH 8.0, 150 mM NaCl, 0.02% sodium azide, 100 µg/mL phenylmethylsulfonyl fluoride, 1 µg/mL aprotinin, and 1% Triton X-100). Total proteins were extracted and purified with 100 µL of chloroform and 400 µL of methanol. Protein samples were boiled for 3 min and subjected to electrophoresis on 8% polyacrylamide gels and then transblotted to nitrocellulose membranes (Bio-Rad Laboratories). Blots were incubated with rabbit polyclonal antibody for TGF-β1 (Proteintech Group, Rosemont, IL, USA) diluted 1:200, USF1 (Alexa-594 Goat anti-Rabbit IgG (A11072, Invitrogen)) diluted 1:500, rabbit anti-GAPDH antibody (ab9486, abcam) diluted 1:2000 as a cytoplasm control, and Mouse Anti-TATA binding protein (TBP, ab51841, abcam) diluted 1:2000 as a nucleus control in 5% non-fat milk in TBST solution (10 mM Tris·HCl, pH 8.0, 150 mM NaCl, and 0.05% Tween 20) for 3 h at room temperature. The membrane was incubated with horseradish peroxidase-labeled secondary antibody (Sigma-Aldrich) for 1 h at room temperature and then washed with TBST once for 15 min and then more four times for 5 min. Immune complexes on the membrane were detected by the enhanced chemiluminescence method (Amersham Pharmacia Biotech, Little Chalfont, Buckinghamshire, UK).

### 4.8. Ethics and Animals

This investigation conformed with the Guide for the Care and Use of Laboratory Animals published by the US National Institutes of Health (NIH Publication No. 85-23, 1996). This study was approved by the ethics committee of Nihon University School of Medicine (approval no. AP18MED056-1, 1 November 2018).

### 4.9. Distribution of FITC-Labeled USF1 PI Polyamide in Kidney

USF1 PI polyamide-3 was labeled with FITC (Sigma-Aldrich, St. Louis, MO, USA). To evaluate the distribution of PI polyamide in kidney, we intraperitoneally injected 1 mg of FITC-PI polyamide into Wistar rats. After 1and 3 days, we removed the kidneys from the rats, prepared frozen specimens, and viewed them.

### 4.10. Creation of Diabetic Rats and Treatment with PI Polyamide

Appendix A shows the experimental protocol used to create the diabetic rats and the administrations of USF1 PI polyamide-3. Male Sprague-Dawley (SD) rats purchased from Japan Charles River Laboratories (Yokohama, Japan) weighing 200 to 250 g were given free access to water and regular laboratory chow. Diabetes was induced in rats by a single intraperitoneal injection of STZ (Sigma-Aldrich) at 60 mg/kg body weight) according to a previous report [20]. Blood glucose was measured from tail vein blood using the *o*-toluidine method in non-fasted conditions. Those rats in which blood glucose level 24 h after STZ treatment rose to 300 mg/dL were subjected to further study.

STZ-induced diabetic rats were intraperitoneally injected with 1 mL of 0.1% acetic acid as control rats. Rats were intraperitoneally injected twice a week with 1 mg/kg body weight of USF1 PI polyamide dissolved in 1 mL of 0.1% acetic acid over a 4-month period and then sacrificed to remove their kidneys. Urinary albumin excretion was measured in 24 h urine collected in metabolic cages for 16 weeks.

### 4.11. Measurement of Body Weight, Proteinuria, Blood Sugar and Renal Function in Rats

Blood glucose was measured in whole blood from rats with Quick Autoneo GLU-HK (Shino-Test Corp, Tokyo, Japan). Hemoglobin A1c (HbA1c) was measured in whole blood from rats with RAPIDIA Auto HbA1c-L (Fujirebio Inc., Tokyo, Japan). Serum UN and creatinine were measured by SRL Inc. (Wako, Saitama, Japan). Twenty-four-hour urine was collected in metabolic cages, and urinary albumin and urinary creatinine were also measured by SRL Inc. Urinary protein excretion was determined with a protein assay kit from Bio-Rad (Hercules, CA, USA). Urinary albumin excretion was expressed as albumin/creatinine ratio.

### 4.12. Morphological and Immunocytochemical Analysis in Vivo

In in vivo experiments, 3-mm-thick paraffin sections were stained with HE and Masson’s trichrome. The diameter of the glomerulus was measured under high magnification (×400). The GIS was obtained by the following formula: [(0 × n0) + (1 × n1) + (2 × n2) + (3 × n3) + (4 × n4)]/50. To semi-quantify the tubulointerstitial area, 20 areas of the renal cortex were randomly selected. The percentage of each area that showed sclerofibrotic change was estimated and assigned a score of 0, normal; 1, involvement of <10% of the area; 2, involvement of 10 to 30% of the area; 3, involvement of 30 to 50% of the area; or 4, involvement of >50% of the area. The TIS was calculated as [(0 × n0) + (1 × n1) + (2 × n2) + (3 × n3) + (4 × n4)]/20.

For TGF-β1 staining, deparaffinized 5-μm-thick sections were incubated with rabbit polyclonal antibody for TGF-β1 (Proteintech Group, Rosemont, IL, USA) diluted 1:200 and were incubated with horseradish peroxidase-conjugated anti-biotin labelling solution (ABC Elite Kit, Vector) at 22 °C for 30 min, followed by washing and further incubation with 3,3-diaminobenzidine solution (DAB). The sections were counterstained before being examined under a light microscope.

### 4.13. Statistical Analyses

Values are reported as means ± SE. The Student *t*-test was used to compare unpaired data, and, as a post hoc test, two-way ANOVA with the Bonferroni/Dunn procedure was also used. A *p*-value < 0.05 was considered to indicate statistical significance.

## 5. Conclusions

Treatments with synthetic USF1 PI polyamide improved renal function, urinary albumin expression and degeneration of kidney by suppressing TGF-β1 in kidney from STZ-diabetic rats. The synthetic USF1 PI polyamide could potentially be a feasible and practical medicine in the treatment of diabetic nephropathy.

## Figures and Tables

**Figure 1 ijms-22-04741-f001:**
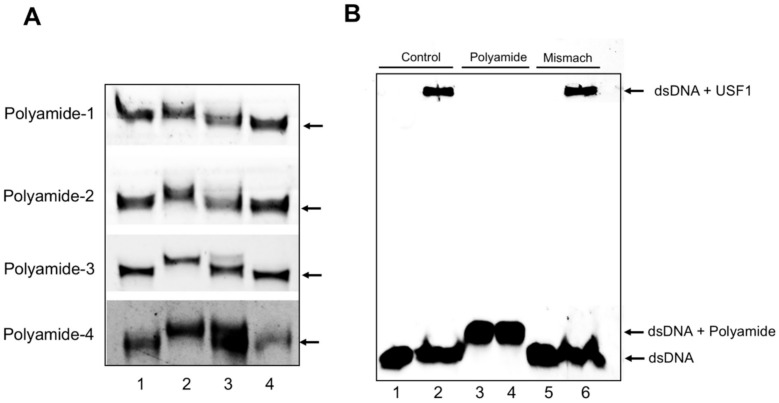
Gel mobility shift and DNA binding assays for upstream stimulatory factor 1 (USF1) pyrrole-imidazole (PI) polyamides. We synthesized 21-mer sense oligonucleotide and antisense oligonucleotide for USF1 PI polyamides (Polyamide-1, -2, -3 and -4). Two oligonucleotides were denatured by heating at 95 °C and then slowly cooled for annealing with each other. (**A**) The double-strand oligonucleotides and the same concentration of PI polyamide dissolved in dimethyl sulfoxide were incubated in binding buffer. Lane 1: FITC-dsDNA (10^−5^ M), Lane 2: FITC-dsDNA (10^−5^ M) + USF1 PI polyamides (10^−5^ M), Lane 3: FITC-dsDNA (10^−5^ M) + excessive dsDNA (10^−3^ M) + USF1 PI polyamide (10^−5^ M), Lane 4: FITC-dsDNA (10^−5^ M) + mismatch PI polyamide (10^−5^ M). Arrows indicate the basal position of the dsDNA. (**B**) FITC-dsDNA was incubated with Polyamide−3 with or without USF1 protein for 30 min at 37 °C and loaded onto a 20% polyacrylamide gel. Lane 1: dsDNA, Lane 2: dsDNA with USF1, Lane 3: DNA with polyamide-3, Lane 4: dsDNA with polyamide-3 and USF1 protein, Lane 5: dsDNA with mismatch, Lane 6: dsDNA with mismatch and USF1 protein.

**Figure 2 ijms-22-04741-f002:**
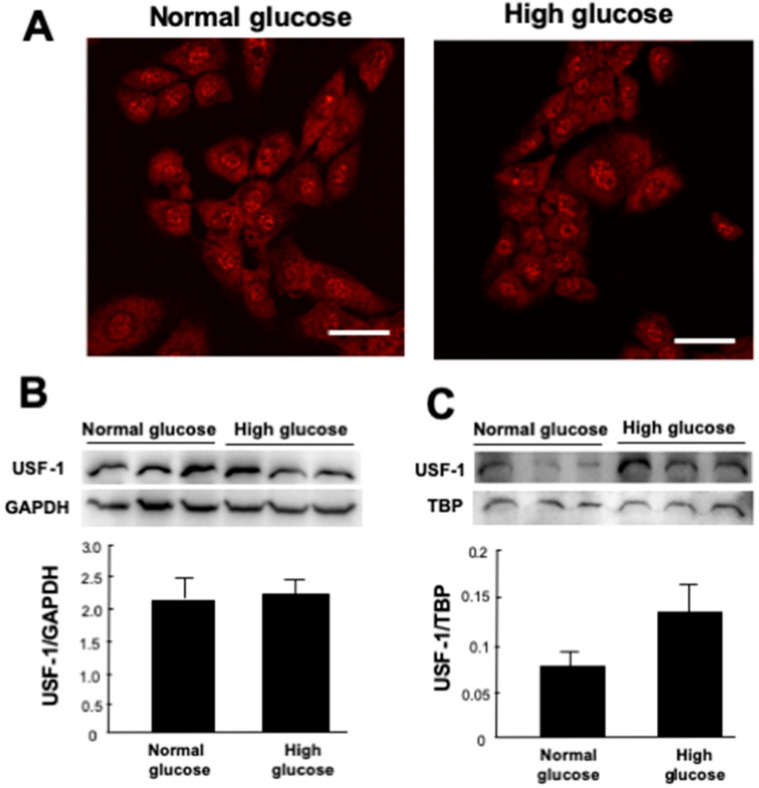
Cellular localization of upstream stimulatory factor (USF1) protein in mesangial cells (MCs) with high-glucose stimulation. (**A**) Immunofluorescence staining of USF1 in MCs. Cells were cultured in normal glucose (5.6 mM) or high glucose (25 mM) in DMEM with 0.5% FBS for 20 h. Then, cells were incubated with rabbit anti-USF1 antibody. Scale bar = 50 μm. Expression of USF1 protein in (**B**) the cytoplasm and (**C**) nucleus of MCs with normal glucose or high-glucose stimulation by Western blot analysis. Data are the mean ± SEM (*n* = 4). GAPDH: a cytoplasm control. TBP: TATA binding protein, a nucleus control.

**Figure 3 ijms-22-04741-f003:**
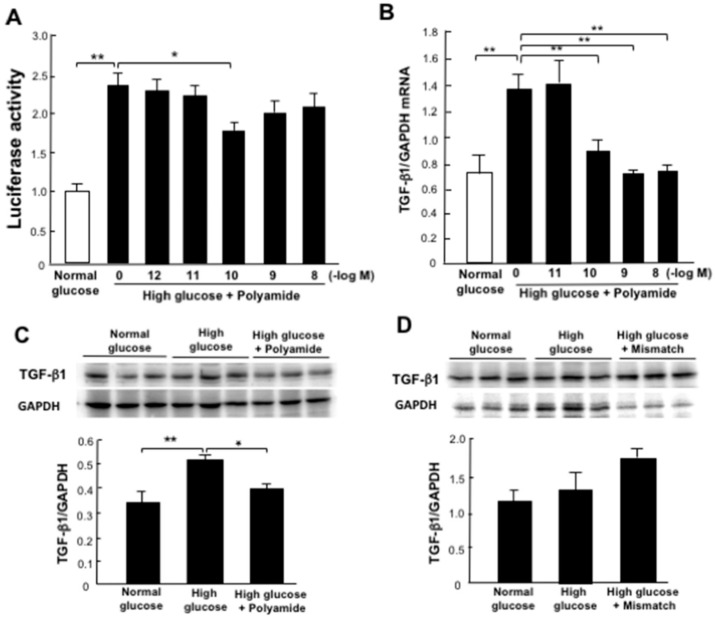
Effects of upstream stimulatory factor 1 (USF1) pyrrole-imidazole (PI) polyamide on transforming growth factor (TGF)-β1 promoter activity and the expression of TGF-β1 with high-glucose stimulation. (**A**) Effects of USF1 PI polyamide on TGF-β1 promoter activity measured by double luciferase activity in HEK293 cells with high-glucose stimulation (*n* = 4). (**B**) Effects of USF1 PI polyamide on the expression of TGF-β1 mRNA in mesangial cells (MCs) with high-glucose stimulation by real-time analysis (*n* = 6). Effects of (**C**) USF1 PI polyamide or (**D**) mismatch polyamide on the expression of TGF-β1 protein in MCs with high-glucose stimulation by Western blot analysis (*n* = 3). Data are the mean ± SEM. * *p* < 0.05, ** *p* < 0.01 between indicated columns.

**Figure 4 ijms-22-04741-f004:**
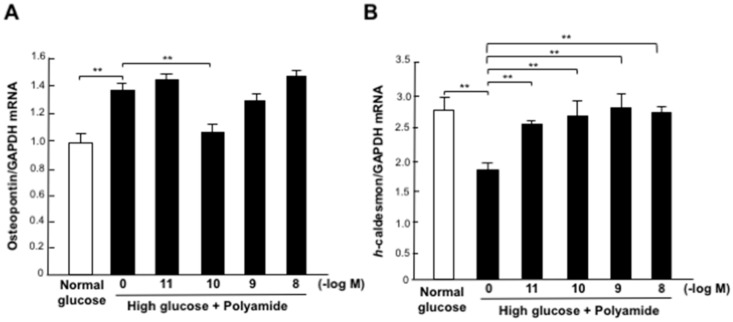
Effects of upstream stimulatory factor 1 (USF1) pyrrole-imidazole (PI) polyamide on the expression of phenotype marker mRNAs in mesangial cells (MCs) with high-glucose stimulation. (**A**) Expression of osteopontin mRNA in MCs treated with high-glucose stimulation and USF1 PI polyamide-3. (**B**) Expression of *h*-caldesmon mRNA in MCs treated with high-glucose stimulation and USF1 PI polyamide-3. Data are the mean ± SEM (*n* = 6). ** *p* < 0.01 between indicated columns.

**Figure 5 ijms-22-04741-f005:**
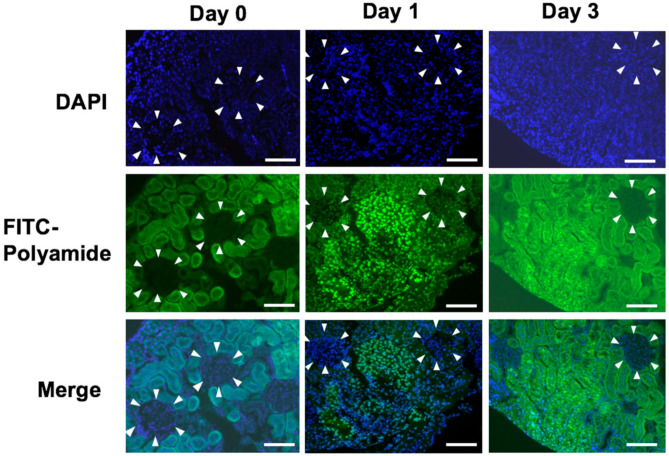
Delivery of USF1 pyrrole-imidazole (PI) polyamide in rat kidney. One milligram of FITC-USF1 PI polyamide-3 was injected intraperitoneally into Wistar rats. After 1 (Day 1) and 3 days (Day 3), the kidneys were removed, and frozen specimens were prepared and viewed. The scale bar indicates 25 μm. The arrow heads distinguish the glomerulus from nephrotubules.

**Figure 6 ijms-22-04741-f006:**
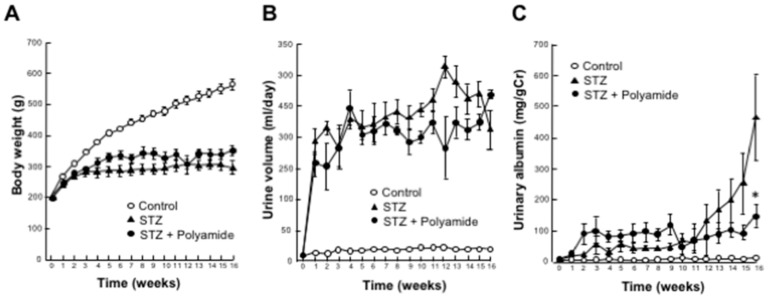
Effects of upstream stimulatory factor 1 (USF1) pyrrole-imidazole (PI) polyamide on (**A**) body weight, (**B**) urine volume and (**C**) urinary albumin excretion in diabetic rats. Diabetes mellitus was induced in rats by single intraperitoneal injection of streptozotocin (STZ). STZ-induced diabetic rats were intraperitoneally injected with 1 mL of 0.1% acetic acid as control rats. Then, 1 mg/kg body weight of USF1 PI polyamide dissolved in 1 mL of 0.1% acetic acid was intraperitonially injected twice a week for 4 months. Urinary albumin was measured in 24 h urine collected in metabolic cages for 4 months. Data are the mean ± SEM (*n* = 6). * *p* < 0.05 between STZ and STZ + Polyamide.

**Figure 7 ijms-22-04741-f007:**
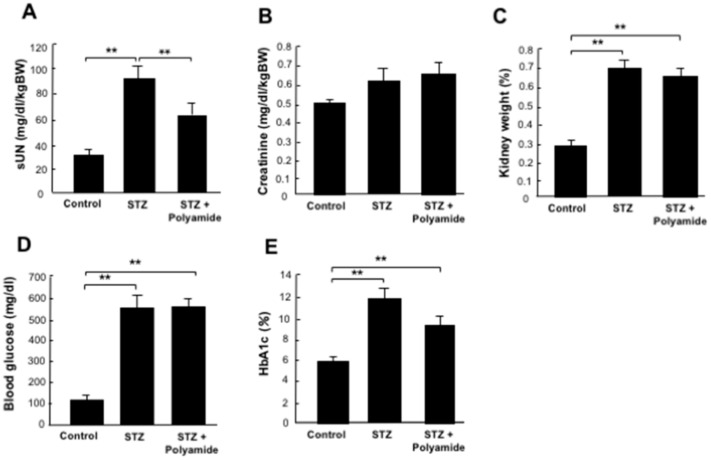
Effects of upstream stimulatory factor 1 (USF1) pyrrole-imidazole (PI) polyamide on (**A**) serum urea nitrogen (sUN), (**B**) serum creatinine, (**C**) kidney weight, (**D**) blood sugar and (**E**) serum HbA1c in diabetic rats. Diabetes mellitus was induced in rats by single intraperitoneal injection of streptozotocin (STZ). STZ-induced diabetic rats were intraperitoneally injected with 1 mL of 0.1% acetic acid as control rats. Then, 1 mg/kg body weight of USF1 PI polyamide dissolved in 1 mL of 0.1% acetic acid was intraperitonially injected twice a week for 4 months. Data are the mean ± SEM (*n* = 6). ** *p* < 0.01 between indicated columns.

**Figure 8 ijms-22-04741-f008:**
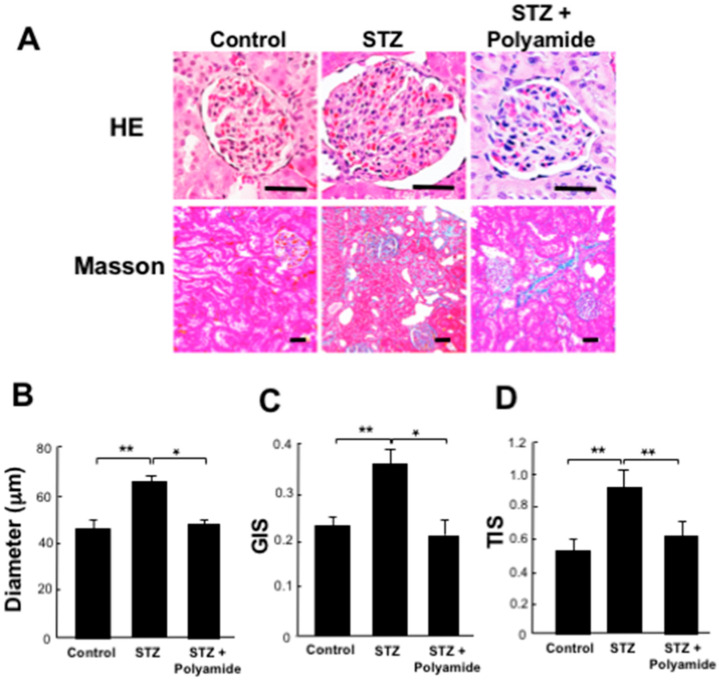
Effects of upstream stimulatory factor 1 (USF1) pyrrole-imidazole (PI) polyamide on kidney degeneration in diabetic rats. Diabetes mellitus was induced in rats by single intraperitoneal injection of streptozotocin (STZ). STZ-induced diabetic rats were intraperitoneally injected with 1 mL of 0.1% acetic acid as control rats. Then, 1 mg/kg body weight of USF1 PI polyamide dissolved in 1 mL of 0.1% acetic acid was intraperitonially injected twice a week for 4 months. (**A**) Three-millimeter-thick paraffin sections were stained with hematoxylin-eosin (HE) and Masson’s trichrome. Under high magnification (×400), (**B**) the diameter of the glomerulus in the renal cortex was measured. (**C**) Glomerular injury score (GIS) and (**D**) tubulointerstitial injury score (TIS) were evaluated. Data are the mean ± SEM (*n* = 8). * *p* < 0.05, ** *p* < 0.01 between indicated columns. The scale bar indicates 25 μm.

**Figure 9 ijms-22-04741-f009:**
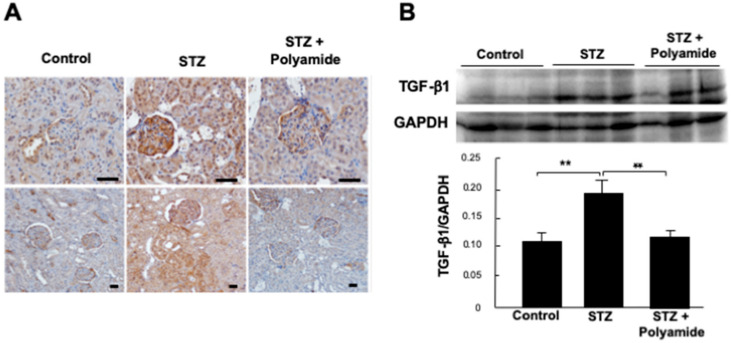
Effects of upstream stimulatory factor 1 (USF1) pyrrole-imidazole (PI) polyamide on the expression of transforming growth factor (TGF)-β1 in streptozotocin (STZ)-diabetic rats. Diabetes mellitus was induced in rats by single intraperitoneal injection of STZ. STZ-induced diabetic rats were intraperitoneally injected with 1 mL of 0.1% acetic acid as control rats. Then, 1 mg/kg body weight of USF1 PI polyamide dissolved in 1 mL of 0.1% acetic acid was intraperitonially injected twice a week for 4 months. (**A**) Immunohistological staining of TGF-β1 in control and STZ-diabetic rats without or with USF1 PI polyamide-3. (**B**) Expression of TGF-β1 in the renal cortex evaluated by Western blot analysis. Data are the mean ± SEM (*n* = 3). ** *p* < 0.01 between indicated columns. The scale bar indicates 25 μm.

**Figure 10 ijms-22-04741-f010:**
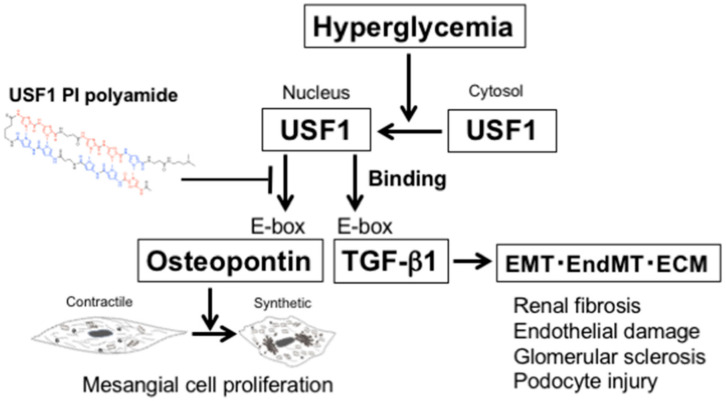
Diagram of the involvement of upstream stimulatory factor 1 (USF1) in the pathogenesis of diabetic nephropathy and the transcriptional inhibition of transforming growth factor (TGF)-β1 and osteopontin promoters by pyrrole-imidazole (PI) polyamide to prevent the binding of USF1 (USF1 PI polyamide). EMT: epithelial–mesenchymal transition, EndMT: endothelial–mesenchymal transition, ECM: extracellular matrix.

**Figure 11 ijms-22-04741-f011:**
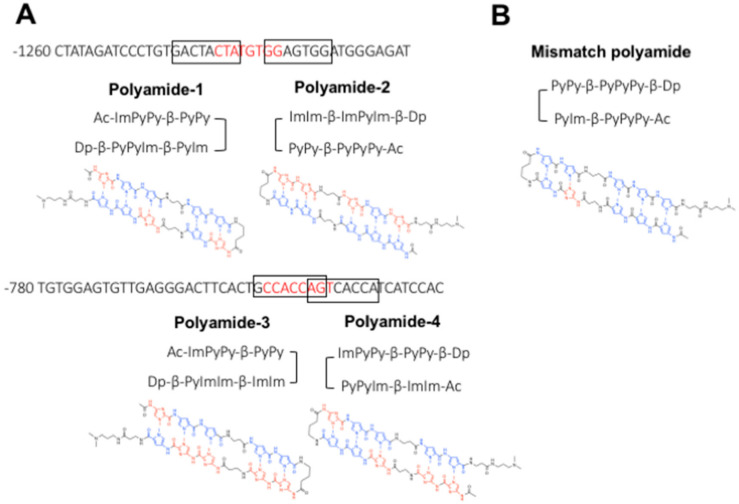
Structure of pyrrole-imidazole (PI) polyamide to prevent the binding of upstream stimulatory factor 1 (USF1) on the transforming growth factor (TGF)-β1 promoter. Polyamides were synthesized by solid-phase methods and were purified by HPLC (0.1% AcOH/CH3CN 0 to 50% linear gradient, 0 to 40 min, 254 nm through a Chemcobond 5-ODS-H column). (**A**) Red capitals are E-box sequences. Boxes indicate the binding site of PI polyamides on the TGF-β1 promoter. Pyrrole structure is indicated by the blue color and imidazole structure is indicated by the red color. (**B**) Structure of mismatch polyamide.

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
