# Peer review of "Transcriptional Suppression of Diabetic Nephropathy with Novel Gene Silencer Pyrrole-Imidazole Polyamides Preventing USF1 Binding to the TGF-β1 Promoter"

_ijms, 2021, doi:10.3390/ijms22094741_

Round 1
Reviewer 1 Report
The use of PI polyamides to inhibit the binding of specific transcription factors to their target promoter sequence is a potentially novel approach for the treatment of a variety of diseases. In this manuscript the USF1 PI polyamide were synthesized and used to presumably inhibit the binding of USF1 to its target sequence in the promoter of TGF-β and inhibit its expression in the kidneys of rats subjected to STZ induced diabetes. The authors demonstrate reductions in some of the indicators of diabetic nephropathy. Based on such alterations the authors suggest that USF1 PI polyamides can be used for the treatment of diabetic nephropathy.
General comments:
The manuscript as it is written requires moderate editing. The data and their presentation require extensive additions and the presentation of data needs to be improved substantially.
Specific comments:
1- In Fig. 1, the EMSA for USF1 PI polyamide is distorted and hard to interpret. The authors need to replaced the gel image with a less distorted one.
2- In Fig. 2A, the pictures need to be larger and a nuclear stain such as DAPI should be used as a counterstain to delineate the nuclei and confirm the nuclear localization of USF1. Authors also need to demonstrate how USF1 PI polyamides, specially USF1 PI polyamide3, affect the nuclear translocation and localization of USF1.
3- In Fig. 2C, why are TBP levels in high glucose (HG) samples elevated in comparison to low glucose (LG) samples? Shouldn't TBP as a loading standard be fairly uniform across the board? Also, the bands are quite irregular!! Have the authors tried desalting the nuclear extracts in order to reduce the distortion of the bands?
4- Does "low glucose" mean normal glucose levels used in tissue culture medium?
5- In all assays using USF1 PI polyamide3, the effect of this PI polyamide on control cells and in control animals will be needed to determine its baseline effect. Also, it will be helpful for authors to show the localization of USF1 PI polyamide3 in cells and kidneys.
6- Fig. 3A. The RT-PCR graphic data needs to be added to the supplement for closer examination.
7- Fig. 3 C and D. The results will be much better presented if the western blots for HG, HG + USF1 PI polyamide3 and HG + Mismatch PI polyamide were run side by side and densitometry was performed on the samples from the same gel. As it is, two separate gels, without knowing the exact exposure time what is shown is not helpful. All densitometry results should be presented on the same graph for easier evaluation. LG and HG densitometry results so different in the two experiments, running a side by side gel will remove this discrepancy.
8- Fig. 4. The RT-PCR graphic data needs to be added to the supplement for closer examination.
9- Fig. 5. The localization of USF1 PI polyamide3 in the kidney needs to be demonstrated. What is the effect of USF1 PI polyamide3 in control mice (Control+polyamine)? This is especially important in part C.
10- Fig. 6. The Control+ USF1 PI polyamide3 data need to be included (This is especially important in part A).
11- Fig. 7A. Need better/higher resolution pictures for Masson Trichrome stain.
12- Fig. 8A. Need larger higher resolution images. In part B, western need to include Control+ USF1 PI polyamide3 lanes.
13- Authors need to include an EMSA showing USF1 binding to its target sequence and the ability of USF1 PI polyamide3 to interfere with this binding. This will confirm the mechanism that is proposed in Fig. 9.
14- Lines 242-245, the statement needs to be reorganized since it is hard to decipher as it is written.
Author Response
Manuscript ID: ijms-1053194
Title: Transcriptional Suppression of Diabetic Nephropathy with Novel Gene Silencer Pyrrole-Imidazole Polyamides Preventing USF1 Binding to TGF-β1 Promoter
Reply to Reviewer 1
Thank you for your comments to our manuscript. Since Editorial Office requires to revise manuscript and send it in 10 days by 20th, in this time I responded your comments and revised manuscript as far as possible without additional experiments using STZ-induced rats that need more 4 months.
Comment 1: In Fig. 1, the EMSA for USF1 PI polyamide is distorted and hard to interpret. The authors need to replaced the gel image with a less distorted one.
Response: In this experiments, importances are gel sift in lane 2 (dsDNA + USF1 PI polyamide) and no sift as lane 3 (dsDNA+excessive dsDNA+ USF1 PI polyamide) and lane 4 (dsDNA + mismatch PI polyamide). These are clearly showed in the gel image. However, I changed contrast to prevent the distores and added arrows indicating basal lposition of the dsDNA.
Comment 2: In Fig. 2A, the pictures need to be larger and a nuclear stain such as DAPI should be used as a counterstain to delineate the nuclei and confirm the nuclear localization of USF1. Authors also need to demonstrate how USF1 PI polyamides, specially USF1 PI polyamide3, affect the nuclear translocation and localization of USF1.
Response: According to your suggestion, we enlarged Figure 2A. Concerning how USF1 PI polyamide3 affect the nuclear translocation and localization of USF1, theoretically the present USF1 PI polyamide3 prevents bindings of USF1 onto TGF-b1 gene in cell nucleus, not prevent the nuclear translocation and localization of USF1, and the suppress the transcription of TGF-b1. That is why we did not show effects of USF1 PI polyamide3 on the nuclear translocation and localization of USF1. Thank you for your understandings.
Comment 3: In Fig. 2C, why are TBP levels in high glucose (HG) samples elevated in comparison to low glucose (LG) samples? Shouldn't TBP as a loading standard be fairly uniform across the board? Also, the bands are quite irregular!! Have the authors tried desalting the nuclear extracts in order to reduce the distortion of the bands?
Response: When attachment of picture of TBP band, gel picture was slightly sloped. We revised Figure 2C by correction of the slope.
Comment 4: Does "low glucose" mean normal glucose levels used in tissue culture medium?
Response: Thank you for your appropriate comment. As your indication, our employed concentration of 5.6 mM glucose is normal glucose, not low glucose. We changed the “low glucose” to “normal glucose” in revised Figures.
Comment 5: In all assays using USF1 PI polyamide3, the effect of this PI polyamide on control cells and in control animals will be needed to determine its baseline effect. Also, it will be helpful for authors to show the localization of USF1 PI polyamide3 in cells and kidneys.
Response: Instead of effects of USF1 PI polyamide-3 on control cells, we showed effects of Mismatch on mesangial cells with high glucose. Mismatch polyamide is identical length and molecular weight to USF1 PI polyamide-3, did not show any effects on expression of TGF-β1 mRNA in mesangial cells with high glucose, indicating that USF1 PI polyamide-3 may not affects the expression of TGF-β1 mRNA in control cells.
Concerning the locarization experiments for PI polyamides, we can not perform additional experiments, because Editorial Office requires to revise manuscript and send it in 10 days. However, we previously demonstrated that FITC-labeled PI polyamides were delivery in almost all nucleuses in nephrotubulus and glomerulus in STZ-diabetic rats (Contribution of TGF-b1 and Effects of Gene Silencer Pyrrole-Imidazole Polyamides Targeting TGF-b1 in Diabetic Nephropathy. Molecules 2020, 25, 950.). Could you refer Figure in Molecules 2020, 25, 950 as below?
Comment 6: Fig. 3A. The RT-PCR graphic data needs to be added to the supplement for closer examination.
Response: In the present experiments, we performed the real-time PCR, not reverse transcription (RT)-PCR. RT-PCR analysis accompanies graphic bands, but the real-time PCR has no graphic data. Thank you for your understanding.
Comment 7: Fig. 3 C and D. The results will be much better presented if the western blots for HG, HG + USF1 PI polyamide3 and HG + Mismatch PI polyamide were run side by side and densitometry was performed on the samples from the same gel. As it is, two separate gels, without knowing the exact exposure time what is shown is not helpful. All densitometry results should be presented on the same graph for easier evaluation. LG and HG densitometry results so different in the two experiments, running a side by side gel will remove this discrepancy.
Response: I understand your comment that high glucose, bands high glucose + USF1 PI polyamide3 and high glucose + Mismatch PI polyamide should be in same electrophoresis. Reason why we performed them in different experiments, in first experiments to examine USF1 PI polyamide-3 on proteinuria, TGF-b1 expression and renal injury in diabetic nephropathy and confirm the effectiveness, and then we examined effects of Mismatch PI polyamide. Thank you for your understandings.
Comment 8: Fig. 4. The RT-PCR graphic data needs to be added to the supplement for closer examination.
Response: In the present experiments, we performed the real-time PCR, not reverse transcription (RT)-PCR. RT-PCR analysis accompanies graphic bands, but the real-time PCR has no graphic data. Thank you for your understandings.
Comment 9: Fig. 5. The localization of USF1 PI polyamide3 in the kidney needs to be demonstrated. What is the effect of USF1 PI polyamide3 in control mice (Control+polyamine)? This is especially important in part C.
Response: We can not perform additional experiments of the locarization of PI polyamides, because Editorial Office requires to revise manuscript in 10 days. However, we previously demonstrated that FITC-labeled PI polyamides were delivery in almost all nucleuses in nephrotubulus and glomerulus in STZ-diabetic rats (Contribution of TGF-b1 and Effects of Gene Silencer Pyrrole-Imidazole Polyamides Targeting TGF-b1 in Diabetic Nephropathy. Molecules 2020, 25, 950.). Could you refer Figure in Molecules 2020, 25, 950? Thank you for your understandings.
Comment 10: Fig. 6. The Control+ USF1 PI polyamide3 data need to be included (This is especially important in part A).
Response: In this reply to comment, we can not show additional experiments of The Control+ USF1 PI polyamide3 data, because experiments to examine USF1 PI polyamide3 on control, non diabetic rats, takes for 4 months. Thank you for your understandings.
Comment 11: Fig. 7A. Need better/higher resolution pictures for Masson Trichrome stain.
Response: I changed to the higher resolution pictures for Masson Trichrome staining.
Comment 12: Fig. 8A. Need larger higher resolution images. In part B, western need to include Control+ USF1 PI polyamide3 lanes.
Response: Figure 8A was enlarged. Again, we can not show additional experiments of The Control+ USF1 PI polyamide3 data, because experiments to examine USF1 PI polyamide3 on control, non diabetic rats, takes for 4 months. Thank you for your understandings.
Comment 13: Authors need to include an EMSA showing USF1 binding to its target sequence and the ability of USF1 PI polyamide3 to interfere with this binding. This will confirm the mechanism that is proposed in Fig. 9.
Response: Thank you for your appropriate comment. Instead of EMSA, we performed the gel mobility shift assay as Figure 1, in which bindings of USF1 PI polyamide-3 on target dsDNA of the USF1 binding sequence, but the prevention of USF1 binding on the target dsDNA with USF1 PI polyamide-3 can not be seen as your indication. If you absolutely require the EMSA to show the prevention of USF1 binding on the target dsDNA with USF1 PI polyamide-3, we will soon perform the EMSA that will take 2 months by the next comments.
Comment 14: Lines 242-245, the statement needs to be reorganized since it is hard to decipher as it is written.
Response: Thank you for your indication. We corrected as followings:
The contractile mesenchymal cells including MCs express contractile phenotype markers such as α-smooth muscle actin, smooth muscle 22α and h-caldesmon, whereas the synthetic mesenchymal cells showing growth with lower contractility expresses synthetic phenotype markers as osteopontin, matrix Gla and l-caldesmon.
Reply to Reviewer 2
Comment 1: Can the authors comment on systemic effects of PA injections? What is the pharmacokinetics? How are they considered safe drugs?
Response: We previously demonstrated that PI polyamide injected intravenously distributed in the kidney and aorta without any drug delivery systems, but did
not distribute considerably in heart or brain in rats (Ref. 7). This statement is added in Introduction in the revised manuscript.
Comment 2: STZ model of DN: What was the mortality? Usually there is high mortality associated with this model?
Response:Mortality in STZ-diabetic rats is renal failure by diabetic nephropathy.
Comment 3: Was insulin given to STZ diabetic rats? If not then why? In real life scenario, patient needs to be on insulin whilst taking any other medication.
Response: Since STZ induces type 1 diabetes mellitus by degenerations of b-cells in pancreas, insulin is absolutely needed, otherwise SGLT2 inhibitor is recently applied for the type 1 diabetes mellitus.
Comment 4: How significant is the effect on other organs where USF downregulation is not needed?
Response: USF1 is upregulated and involuved in hepatoma, breast cancer and glioblastoma et al. (Immunochemical characterization and transacting properties of upstream stimulatory factor isoforms. Viollet, B., Lefrançois-Martinez, A.M., Henrion, A., Kahn, A., Raymondjean, M., Martinez, A. J. Biol. Chem. (1996), Loss of USF transcriptional activity in breast cancer cell lines. Ismail, P.M.,Lu, T., Sawadogo, M. Oncogene (1999)., Salero, E., Giménez, C., Zafra, F. Biochem. J. (2003) Identification of a non-canonical E-box motif as a regulatory element in the proximal promoter region of the apolipoprotein E gene. , Immunochemical characterization and transacting properties of upstream stimulatory factor isoforms. Viollet, B., Lefrançois-Martinez, A.M., Henrion, A., Kahn, A., Raymondjean, M., Martinez, A. J. Biol. Chem. (1996)., Loss of USF transcriptional activity in breast cancer cell lines. Ismail, P.M., Lu, T., Sawadogo, M. Oncogene (1999)., Salero, E., Giménez, C., Zafra, F. Biochem. Identification of a non-canonical E-box motif as a regulatory element in the proximal promoter region of the apolipoprotein E gene.J. (2003).
Thank you
Reviewer 2 Report
In their paper entitled " Transcriptional Suppression of Diabetic
3 Nephropathy with Novel Gene Silencer Pyrrole4 Imidazole Polyamides Preventing USF1 Binding to 5 TGF-β1 Promoter" Okamura et al have described the effects of novel gene silencer polyamides on attenuation of STZ induced nephropathy. Overall, the study is well designed and written. I am especially intrigued by role of PA's as drugs. Here are my comments:
- Can the authors comment on systemic effects of PA injections? What is the pharmacokinetics? How are they considered safe drugs?
- STZ model of DN: What was the mortality? Usually there is high mortality associated with this model?
- Was insulin given to STZ diabetic rats? If not then why? In real life scenario, patient needs to be on insulin whilst taking any other medication.
- How significant is the effect on other organs where USF downregulation is not needed?
Author Response
Manuscript ID: ijms-1053194
Title: Transcriptional Suppression of Diabetic Nephropathy with Novel Gene Silencer Pyrrole-Imidazole Polyamides Preventing USF1 Binding to TGF-β1 Promoter
Reply to Reviewer 2
Comment 1: Can the authors comment on systemic effects of PA injections? What is the pharmacokinetics? How are they considered safe drugs?
Response: We previously demonstrated that PI polyamide injected intravenously distributed in the kidney and aorta without any drug delivery systems, but did
not distribute considerably in heart or brain in rats (Ref. 7). This statement is added in Introduction in the revised manuscript.
Comment 2: STZ model of DN: What was the mortality? Usually there is high mortality associated with this model?
Response:Mortality in STZ-diabetic rats is renal failure by diabetic nephropathy.
Comment 3: Was insulin given to STZ diabetic rats? If not then why? In real life scenario, patient needs to be on insulin whilst taking any other medication.
Response: Since STZ induces type 1 diabetes mellitus by degenerations of b-cells in pancreas, insulin is absolutely needed, otherwise SGLT2 inhibitor is recently applied for the type 1 diabetes mellitus.
Comment 4: How significant is the effect on other organs where USF downregulation is not needed?
Response: USF1 is upregulated and involuved in hepatoma, breast cancer and glioblastoma et al. (Immunochemical characterization and transacting properties of upstream stimulatory factor isoforms. Viollet, B., Lefrançois-Martinez, A.M., Henrion, A., Kahn, A., Raymondjean, M., Martinez, A. J. Biol. Chem. (1996), Loss of USF transcriptional activity in breast cancer cell lines. Ismail, P.M.,Lu, T., Sawadogo, M. Oncogene (1999).
Salero, E., Giménez, C., Zafra, F. Biochem. J. (2003) Identification of a non-canonical E-box motif as a regulatory element in the proximal promoter region of the apolipoprotein E gene. , Immunochemical characterization and transacting properties of upstream stimulatory factor isoforms. Viollet, B., Lefrançois-Martinez, A.M., Henrion, A., Kahn, A., Raymondjean, M., Martinez, A. J. Biol. Chem. (1996)., Loss of USF transcriptional activity in breast cancer cell lines. Ismail, P.M., Lu, T., Sawadogo, M. Oncogene (1999)., Salero, E., Giménez, C., Zafra, F. Biochem. Identification of a non-canonical E-box motif as a regulatory element in the proximal promoter region of the apolipoprotein E gene.J. (2003).
Thank you
Round 2
Reviewer 1 Report
Determining the localization of USF1 PI polyamide-3 (comment 9) and its ability to compete for the binding if USF1 to its target sequence (comment 13) are two major points that needed to be addressed. These are important experiments that improve the quality of the manuscript; however, they require more time than the 10 days resubmission deadline set by the journal (according to authors). If the 10 day resubmission deadline is firm then I would accept the manuscript for publication since authors have addressed the rest of my concerns. If the journal and the editor are willing to extend the resubmission deadline (90 days is normal resubmission deadline for many major journals), then I think addressing the localization of the USF1 PI polyamide-3 and its ability to compete for binding of USF1 to its target need to be included.
Author Response
Replies to comments from Reviewer 1
Comments
Determining the localization of USF1 PI polyamide-3 (comment 9) and its ability to compete for the binding if USF1 to its target sequence (comment 13) are two major points that needed to be addressed. These are important experiments that improve the quality of the manuscript; however, they require more time than the 10 days resubmission deadline set by the journal (according to authors). If the 10 day resubmission deadline is firm then I would accept the manuscript for publication since authors have addressed the rest of my concerns. If the journal and the editor are willing to extend the resubmission deadline (90 days is normal resubmission deadline for many major journals), then I think addressing the localization of the USF1 PI polyamide-3 and its ability to compete for binding of USF1 to its target need to be included.
Responses
According to Reviewer 1’s comments, we performed additional experiments to
examine the delivery of FITC labeled-PI polyamide in rat kidney as Figure 5. We demonstrate that FITC-labeled USF1 PI polyamide-3 was certainly distributed into the nucleus of the nephron tubule, and the glomerulus after peritorial injections of Polyamide-3.
In addition, we also performed an EMSA showing USF1 binding to its target sequence and the ability of USF1 PI polyamide-3 to interfere with this binding of USF1. Polyamide-3 completely inhibited USF1 binding to target dsDNA.
Thank you

Round 3
Reviewer 1 Report
The manuscript is much improved and the additional data clarify many of the points that were not clear.
Question:
How do authors reconcile the reduction in fibrosis, normal GIS and TIS with increased kidney weight in STZ-USF1-PI polyamide-3 rats? What is the increase in kidney weight due to?
Minor points:
Lines 191: "Treatment with USF1 PI polyamide-3 significantly (p < 0.05) decreased the increase in urinary albumin excretion in the STZ-diabetic rats" the underlined portion can be replaced with "... significantly reduced the excretion of albumin in diabetic rats"
Line 202: If serum urea nitrogen is defined as (UN) for consistency use either serum UN or define the term as sUN.
Line 225 and 226: replace "significantly inhibited increases" with "decreased."
Fig. 9 legend. Abbreviations are already in the text.
Line 305: please replace "created" with "generated."
Line 315: Please replace "significantly decreased the increase ..." with significantly attenuated the increase...."
Line 323: Replace "weaken the enhanced staining of TGF-β1" with "reduces the staining of TGF-β1."
Line 524 and 525: "urinary albumin expression" should be "urinary albumin excretion"
Line 525: "kidney from STZ-diabetic rats" should be "kidneys of STZ-diabetic rats."
Author Response
Manuscript ID: ijms-1053194-R3
Title: Transcriptional Suppression of Diabetic Nephropathy with Novel Gene Silencer Pyrrole-Imidazole Polyamides Preventing USF1 Binding to TGF-β1 Promoter
Reply to Reviewer 1 Round 3
Question: How do authors reconcile the reduction in fibrosis, normal GIS and TIS with increased kidney weight in STZ-USF1-PI polyamide-3 rats? What is the increase in kidney weight due to?
Response: Increases in kidney weights in STZ-DM rats may be associated with not only glomerular enlargement and intestinal fibrosis, but also degeneration of whole kidney structure such as collagen tissues and basal membrane of Bowman’s capsula etc. Thus it is possible that administration of PI polyamide-3 did not significantly reduce the kidney weight. However these considerations are just speculation, not added in the revised manuscript.
Minor points:
Comment: Lines 191: "Treatment with USF1 PI polyamide-3 significantly (p < 0.05) decreased the increase in urinary albumin excretion in the STZ-diabetic rats" the underlined portion can be replaced with "... significantly reduced the excretion of albumin in diabetic rats"
Response: replaced
Comment: Line 202: If serum urea nitrogen is defined as (UN) for consistency use either serum UN or define the term as sUN.
Response: changed to “sUN”
Comment: Line 225 and 226: replace "significantly inhibited increases" with "decreased."
Response: changed to “decreased”
Comment: Fig. 9 legend. Abbreviations are already in the text.
Response: we do not change
Comment: Line 305: please replace "created" with "generated."
Response: changed to "generated"
Comment: Line 315: Please replace "significantly decreased the increase ..." with significantly attenuated the increase...."
Response: changed to “significantly attenuated the increas"
Comment: Line 323: Replace "weaken the enhanced staining of TGF-β1" with "reduces the staining of TGF-β1."
Response: reduces the staining
Comment: Line 524 and 525: "urinary albumin expression" should be "urinary albumin excretion"
Response: changed to "urinary albumin excretion"
Comment: Line 525: "kidney from STZ-diabetic rats" should be "kidneys of STZ-diabetic rats."
Response: changed to “kidneys of STZ-diabetic rats”
